# The role of sex and femininity in preferences for unfamiliar infants among Chinese adults

Fangyuan Ding[1,2☯], Gang Cheng[3☯], Yuncheng Jia[1,2], Wen Zhang[3], Nan Lin[3], Dajun Zhang[1,2]*, Wenjing Mo[1,2]

1 Faculty of Psychology, Southwest University, Beibei, Chongqing, China, 2 Center for Mental Health Education, Southwest University, Beibei, Chongqing, China, 3 School of Psychology, Guizhou Normal University, Guiyang, Guizhou, China

☯ These authors contributed equally to this work.
* zhangdj@swu.edu.cn

**Data Availability Statement:** All relevant data are within the manuscript and its Supporting information files.

## Abstract

Guided by parental investment theory and social role theory, this study aimed to understand current contradictory results regarding sex differences in response to infant faces by considering the effect of gender role orientation. We recruited 300 adults in China and asked them to complete an Interest in Infants questionnaire and a Bem Sex Role Inventory and then administered a behavioral assessment that used unfamiliar infant faces with varying expressions (laughing, neutral, and crying) as stimuli to gauge three components of motivation towards infants (i.e., liking, representational responding, and evoked responding). The results demonstrated that sex differences emerged only in self-reported interest in infants, but no difference was found between the sexes in terms of their hedonic reactions to infant faces. Furthermore, femininity was found to correlate with preferences for infants in both verbal and visual tests, but significant interactive effects of feminine traits and sex were found only in the behavioral test. The findings indicated that men's responses to infants were influenced more by their feminine traits than were women's responses, potentially explaining the greater extent to which paternal (vs. maternal) investment is facultative.

## Introduction

According to parental investment theory [1], the relative proportion of parental investment varies between males and females. For most mammals, female investment in parenting is heavier than male investment, albeit with some exceptions [2, 3]. Among humans, men assume lower levels of minimum parental investment than women do [4]. Specifically, women are forced to bear the cost of fertilization, gestation and even lactation. In contrast, the minimum physiological obligation of men is merely the contribution of sperm, which is considerably less than the obligations of women. Furthermore, lactation could last several years for ancestral women, which made it more difficult for women than for men to reproduce and invest in additional offspring [5]. Thus, male parental investment is presumably more malleable than female investment.

**Funding:** This work was supported by the National
Natural Science Foundation of China (NO.
31671149) to DZ; Chongqing postgraduate
innovative research project (NO. CYS17052) to FD;
the Special Project for Academic Novice Cultivation
and Innovative Exploration (NO.
QianKeQuanPingTaiRenCai[2017]5726-16) to GC,
supported by Guizhou Provincial Science and
Technology Foundation (GPSTF); and PhD early
development program of Guizhou Normal
University (2017) to GC, supported by the Guizhou
Normal University (GZNU). The funders had no role
in study design, data collection and analysis,
decision to publish, or preparation of the
manuscript.

**Competing interests:** The authors have declared
that no competing interests exist.

Some cross-cultural studies in humans verify this tendency by showing that mothers contribute higher parental effort [6, 7], but wide variability is documented in male parental investment [8]. For example, fathers in hunter-gatherer societies (e.g., Aka foragers in Central Africa) interact with children more often than fathers in most Western societies and pastoral societies, while in many farming societies, fathers never hold their infants at all [6]. This interesting pattern found in humans, called male facultative investment, has received considerable attention from researchers. Cultural differences, such as the social constraints of polygamy and the operational sex ratio, have been thought to contribute to this pattern [9], while sex differences in parental investment are not consistently stable, even in similar cultures.

Various studies using the self-reported method have also revealed substantial sex differences among adults. For instance, emerging women were found to attach more personal salience to the parental role, were more likely to state intentions to assume a parental role in the future, and had more expectations regarding the parental role than men [10]. Women consistently reported more interest in infants than men did in numerous studies of both undergraduates and parents [11–14].

The findings are less clear in recent studies that have used images of infant faces as hedonic stimuli and employed novel methods to assess preferences for infants (i.e., physiological and behavioral reactions), as these have found conflicting results. Specifically, in some neuroimaging studies, males and females have been found to demonstrate different neural patterns in response to infant faces [15], while in other studies, no difference was found in the brain activations of males and females in response to infant faces [16].

Similar inconsistencies have also emerged in regard to adults' behavioral responses to infant faces. In terms of infant attractiveness ratings, some studies have found that women give higher ratings than men [17–19], while other studies have failed to replicate these findings [16, 20–22]. Regarding visual preference for infant faces, in some studies, women have been found to exert more effort to prolong their viewing time than men [12, 23], whereas in other studies, no difference was found between the sexes [14, 17–20, 22]. Furthermore, no sex difference was found in relation to the motivation to care for infants [21] or the likelihood to adopt [20].

Although most of the above studies did not directly involve parents' investment in offspring, self-reported interest in infants and preferences for infant faces are useful for evaluating both childless adults and parents' investment for two reasons. First, a unique cooperative parenting system has evolved in humans and has been sustained in part by the interest in infants of non-kin adults [24]. Studies in human and nonhuman primates have indicated that interest in infants is "a developmental adaptation to facilitate the acquisition of parenting skills", which are important for the survival of offspring [25]. Second, recognizing infant cues and responding to infant needs are important parts of parenting. Hence, the response to infant faces is believed to play an important role in actual parenting behavior [26].

More importantly, it is not difficult to see from the above studies that studies using questionnaires have consistently found sex differences in interest in infants, while the findings of studies using infant pictures as stimuli have been contradictory. One possible explanation is that the questionnaire method is vulnerable to social desirability, which may magnify the differences between the sexes, although reactions to infant faces are less influenced by social desirability. Furthermore, it is possible that some within-sex variation, such as developmental correlates, may modify the differences between the sexes.

The social roles of women and men have changed dramatically in recent decades [27], and changes in social norms have been thought to stem from men and women placing a high value on both work and family roles [28]. Based on social role theory [29, 30], these dramatic shifts in social roles should result in corresponding shifts in the traits and behaviors that are considered appropriate for contemporary adults. To the extent that women and men occupy roles

involving domestic activities or economically productive activities, the associated skills, values, and motives are incorporated into their gender roles. Gender roles, along with the specific roles occupied by men and women (e.g., provider, homemaker), guide social behavior, including infant caregiving [30].

Based on this view, plentiful research on adults has revealed a substantial link between feminine internalization and parenting with regard to self-assessed parental behaviors, emotions, and cognitions. For instance, research using the self-report method has demonstrated that adults who have internalized more feminine traits are more accessible to their children [31]; report greater intention to parent, expectations of parenting, and appreciation for the salience of parenting [10]; demonstrate a stronger nonparent desire to have children [32]; and rate higher on the likelihood to adopt for neutral and smiling children [20]. Thus, consideration of the internalized roles adopted by the sexes (i.e., gender roles) may help us resolve the existing controversy regarding motivational reactions to infant faces. In particular, most previous studies of the sex differences in response to unfamiliar infants have not considered gender roles.

In addition to the failure to consider gender roles, two deficiencies of previous studies about reactions to infant faces should be noted. First, parent-child interactions primarily consist of expressions and voices [33], but most existing studies using the behavioral paradigm have used only neutral infant faces as the hedonic stimuli to investigate the differences between the sexes [11, 12, 17–19, 21]. In fact, smiling and crying are known to convey a child's emotional state [34, 35] and signal the need for certain resources from potential caregivers [36]. Empirical studies have also found that infants' and children's facial expressions have effects on adults' behavioral responses and brain activity [20, 37–39]. Thus, it is important to note the impact of infant emotions on adults' interest in nurturing.

Second, previous studies have assessed only participants' liking (conscious pleasure) and evoked responding (wanting and making effort to extend the viewing time) to infants [11, 12, 17–19, 22, 23]. Berridge and Robinson [40] expanded the motivational system in their review to also include representational responding (wanting and conscious desires based on cognitive expectations), and these three components have different neural substrates. More directly, if an individual interacts with a baby, liking represents how much pleasure the individual feels when he or she first sees the baby, evoked responding indicates the individual's willingness to spend more time interacting with the baby, and representational responding signifies the individual's willingness to interact with the baby in the future when separated from that baby. Thus, these three components may account for different proportions of parental investment.

Based on the above considerations, to make our study comparable to previous studies, we employed a questionnaire to gauge interest in infants and a behavioral paradigm to measure liking and evoked responding in relation to infants; these measures were adopted in most previous studies [12, 17–19, 22, 23]. We further broadened the generalizability of our findings by using infant faces with different expressions as stimuli and assessed representational responding together with liking and evoked responding. We further expanded the study by considering the many different ethnic groups in China, which usually have different policies and customs. For example, minority groups (i.e., non-Han), which account for 8.5% of the population, are permitted to have two or more children, whereas Han, who make up 91.5% of the population, were only allowed to have one child prior to 2016. When the two-child policy was officially implemented on January 1, 2016, all Chinese were permitted to have two children. For ethnic minorities, local fertility policies are formulated in light of local realities by the local government of the autonomous region. Therefore, considering the potential impact of different fertility policies on these ethnic groups, ethnic minorities were included as a control variable in this study.

In sum, this study used both a self-reported method and a behavioral paradigm to measure preferences for infants and explored the effects of sex and gender role orientation among Chinese adults. Following two existing studies among Chinese that found sex differences in self-reported interest in infants but no sex differences in behavioral responses and brain activity upon viewing infant faces [14, 16], we hypothesized similar results in our study. Beyond that, the current study aimed to explore the following two main research questions: a) To what extent do gender and gender role orientation among childless adults influence their infant preferences? b) Are these differences or effects influenced by infant facial expressions?

## Method

### Participants

With the assistance of community workers and staff in the Civil Affairs Bureau of Guizhou Province, China, we recruited 300 healthy childless adults (154 women, 146 men) to participate in the study based on three inclusion criteria: a) no history of mental illness; b) no children; and c) age between 18 and 40 years old. Of these adults, 34.3% were unmarried, 51.3% were of Han ethnicity, and 89.0% were employed (85.1% of women; 93.2% of men). The participants ranged in age from 18 to 40 years old (M = 26.090, SD = 3.321). The sample size broken down by sex, marital status and ethnicity is reported in the (S1 Table). Each participant received 50 yuan as compensation for her or his anonymous participation. This study was approved by the Ethics Committee of Southwest University (No. 2014179).

### Procedure

After briefly introducing the study, we obtained written informed consent from the participants. They then completed the hard copy of the self-administered questionnaires, which included demographic questions, the Bem Sex Role Inventory (BSRI), and the Interests in Infants questionnaire. Finally, the participants were instructed to complete a computer task (described below). The entire process took approximately 30 minutes. We did not collect participants' names or personal contact information to ensure anonymity.

### Measures

**Gender role orientation.**   The Chinese version of the BSRI, translated by Yang et al. [41], was adopted to gauge participants' masculine and feminine characteristics. The BSRI is a reliable and valid measure proven in Chinese samples [42] comprising 60 self-report items scored on a Likert scale ranging from 1 "never or almost never true" to 7 "always or almost always true." It contains three 20-item subscales: a masculinity scale (e.g., ambitious and aggressive), a femininity scale (e.g., gentle and affectionate), and a gender-neutral scale (e.g., helpful and happy). The Cronbach's α of the study sample was 0.888 for masculinity and 0.779 for femininity. In this study, average scores for masculinity and femininity were calculated.

**Interest in infants.**   In the present study, the Chinese version of the Interest in Infants questionnaire was employed [14]. This 10-item questionnaire, which has been demonstrated to be reliable and valid [12], asks, "If you were at a party and there was a baby in the room that you did not know, what would you most likely do?" Ten different types of interactions with the baby are then listed (e.g., go over to see the baby at least once). Responses are given on a 6-point Likert scale from 1 (not at all likely) to 6 (very likely). Items indicating avoidance of the infant are reverse-coded. Thus, a higher score indicates higher interest in infants. The Cronbach's α in this study was 0.846.

**(A) Self-reported liking and representational responding**

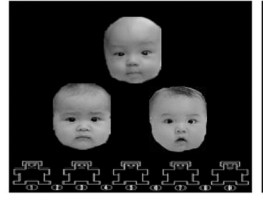

| | Screen blank<br><br>3 seconds | Do you want to<br>see this slide again<br>later?<br><br>2 seconds | Screen blank<br><br>2 seconds<br><br>before new slide<br>presented | |
|---|---|---|---|---|

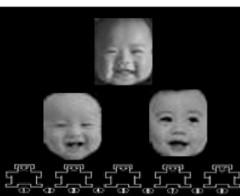

Self-Reported Liking
Unlimited time

Representationa
l

**(B) Evoked responding**

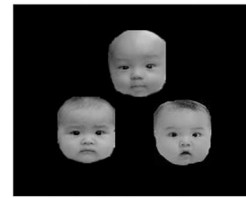

| | Screen blank | |
|---|---|---|

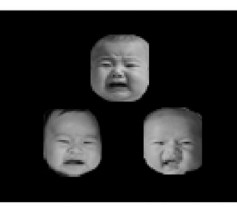

5 seconds ± participant adjustment
to maximum time of 10 seconds

**Fig 1. Experimental procedure.** Note. These infact faces were reprinted from the Chinese Infact Affective Face Picture System (CIAFS) under a CC BY license, with permission from Dr. Gang Cheng, original copyright [2015].

**Motivational values towards infants.** Facial stimuli, consisting of 24 slides of infants' faces (8 laughing, 8 neutral, and 8 crying), were used. All slides contained three grayscale normalized face images that were matched for size and luminosity. These faces were standardized and taken from the Chinese Infant Affective Face Picture System [43].

Motivational values towards infants were assessed using a computer task similar to that used in the studies of Cheng et al. [14] and Ding et al. [39]. This task contains three sections to evaluate different components of the motivational system: liking (hedonic experience), representational responding (cognitive salience and wanting), and evoked responding (incentive salience and wanting). Fig 1 shows an outline of the experimental procedure.

To measure liking, the participants rated the extent of the pleasure experienced from each slide on the 9-point self-assessment manikin, ranging from 1 (extremely unpleasurable) to 9 (extremely pleasurable), with 5 (uncertain) as the midpoint [44].

Following the rating, representational responding (wanting) was measured. Before beginning the assessment, the participants were informed that they would view a slideshow of some of the slides they had previously rated. If they wanted to see a slide again, they could press the "n" and "m" keys alternately, or they could press the "x" and "z" keys if they did not want to see a slide again. The more "n" and "m" keys they pressed, the more likely they were to see the face again and vice versa for the "x" and "z" keys. If they were indifferent to whether a slide showed again, they could choose not to press any keys. This procedure required memory representations due to the absence of stimuli during the response.

Evoked responding ("wanting") was measured by exposing participants to a sustaining stimulus. Eighteen of the previously viewed slides (six infants for each expression) were used in this part of the study. Similar to representational responding, the more "n" and "m" keys the participants pressed, the longer the viewing time was, while the "x" and "z" keys were used to

**Table 1. Summary of variables.**

| Name of Variables | M±SD | | Skewness | Kurtosis |
|---|---|---|---|---|
| | Female(N = 154) | Male(N = 146) | | |
| Age | 25.195±2.997 | 27.034±3.395 | 0.602 | 1.228 |
| Femininity | 4.899±0.685 | 4.784±0.653 | 0.128 | 0.054 |
| Masculinity | 4.443±0.867 | 5.194±0.653 | 0.625 | 0.517 |
| Interest | 45.571±10.546 | 42.082±9.206 | 0.399 | 0.154 |
| Liking | 6.071±1.309 | 6.309±1.393 | 0.130 | 0.288 |
| Representational | 37.543±38.222 | 45.192±55.175 | 0.595 | 0.254 |
| Evoked | 145.615±106.698 | 166.578±126.387 | 0.227 | 0.841 |

shorten the viewing time. The slides were presented randomly, and the participants were told that no matter what keys they pressed, the time for this part was fixed.

Liking for infants was measured by calculating the average score for the liking of all infant faces. For representational responding, the total number of presses to seek desired images was subtracted from the total number of presses to avoid undesired images. The same calculation was used to measure evoked responses to infants.

## Statistical procedure

Following the research questions, we first summarized the variables used and conducted a Pearson correlation analysis and ANOVA in SPSS 22.0. Then, hierarchical regression was employed with SPSS 22.0 to verify the sex differences and the interaction between sex and gender role orientation. Finally, the multilevel models were estimated to verify the effect of facial expression with hierarchical linear and nonlinear modeling (HLM) software.

## Results

### Preliminary analysis

**Descriptive statistics and correlational analysis.** All the continuous variables used in the subsequent analysis are summarized in Table 1. The correlations between continuous variables are presented in Table 2. The results demonstrate that both femininity and masculinity are congruously positively correlated with interest in infants and three domains of motivation

**Table 2. Correlations between variables.**

| | 1 | 2 | 3 | 4 | 5 | 6 |
|---|---|---|---|---|---|---|
| 1. Age | - | - | - | - | - | - |
| 2. Femininity | 0.047 | - | - | - | - | - |
| 3. Masculinity | 0.241*** | 0.404*** | - | - | - | - |
| 4. Interests | 0.085 | 0.304*** | 0.149* | - | - | - |
| 5. Liking | 0.062 | 0.240*** | 0.215** | 0.281*** | - | - |
| 6. Representational | 0.094 | 0.156** | 0.136* | 0.133* | 0.606*** | - |
| 7. Evoked | 0.048 | 0.195** | 0.182** | 0.183** | 0.502*** | 0.630*** |

Note.

* $p < .05$,

** $p < .01$,

*** $p < .001$.

for infants (i.e., self-reported liking, representational responding, and evoked responding) ($r = .136–.304$, $p < .05$).

**ANOVA results.** A 2(sex)×2(marital status)×4(ethnicity: Han, Miao, Dong, Others) ANOVA was conducted to test the effect of the categorical variables. We found that males had higher masculinity and less interest in infants than females, while for marital status and ethnicity, no significant main effect or interaction effect was detected. More details of the ANOVA are provided in the (S2 Table).

Because ethnicity had no main effect on the dependent variables and minority groups in China are usually permitted to have two children, we recoded ethnicity into a dummy variable (0 = Han, 1 = Minority) as a confounding variable in the subsequent analysis.

## Hierarchical regression

A hierarchical regression analysis was conducted to identify the unique role of sex, femininity, masculinity, and the interaction between gender roles and sex in self-reported interest in infants and motivation towards infant faces irrespective of infant facial expressions. Given the potential effects of age, ethnicity, and marital status, they were entered in the first step, sex was entered in the second step, femininity was entered in the third step, masculinity was entered in the fourth step, and interactions between femininity and sex and between masculinity and sex were entered in the fifth step. Before conducting the analysis, femininity and masculinity were centered, and interaction was generalized with centered gender roles [45].

After controlling for age, ethnicity, and marital status, only femininity consistently positively predicted the interest in infants ($\beta = 0.260$, $p < .01$) and motivational values ($\beta = 0.147–0.219$, $p < .05$) (see Tables 3–6). When the interaction of gender roles and sex was added to the equation, only interest in infants was still significantly predicted by femininity ($\beta = 0.193$, $p < .05$), and the interactions were nonsignificant. Regarding responding to infant faces, we found an accordant positive significant interaction between femininity and sex.

Process 3.0 (Model 1) was used to break down the interaction effects. The results showed that femininity did not predict the motivational values of women ($t = -0.114–0.548$, $p > .05$),

**Table 3. Hierarchical regression analyses of interest in infants.**

| Model | β in Step 1 | β in Step 2 | β in Step 3 | β in Step 4 | β in Step 5 |
|---|---|---|---|---|---|
| Age | 0.062 | 0.121* | 0.104 | 0.093 | 0.093 |
| Ethnicity | 0.114* | 0.112* | 0.090 | 0.087 | 0.089 |
| MS | 0.162** | 0.155** | 0.134* | 0.128* | 0.130* |
| Sex | | -0.207*** | -0.180** | -0.230*** | -0.227*** |
| Fem | | | 0.260*** | 0.210*** | 0.193* |
| Mas | | | | 0.117 | 0.137 |
| Fem × Sex | | | | | 0.029 |
| Mas × Sex | | | | | -0.033 |
| $R^2$ | 0.053** | 0.093*** | 0.158*** | 0.167*** | 0.167*** |
| $\triangle R^2$ | | 0.039*** | 0.066*** | 0.008 | 0.001 |

Note: Standardized coefficients are reported. MS = Marital status (Married = 1, Unmarried = 0), Fem = Femininity, Mas = Masculinity, Ethnicity (Minority = 1, Han = 0), Sex (Men = 1, Women = 0),

* $p < .05$,

** $p < .01$,

*** $p < .001$.

The same pertains to the following tables.

**Table 4. Hierarchical regression analysis of liking.**

| Model | β in Step 1 | β in Step 2 | β in Step 3 | β in Step 4 | β in Step 5 |
|---|---|---|---|---|---|
| Age | 0.036 | -0.011 | -0.003 | -0.011 | -0.009 |
| Ethnicity | 0.140* | 0.141* | 0.122* | 0.120* | 0.134* |
| MS | 0.185** | 0.188** | 0.170** | 0.166** | 0.174** |
| Sex | | 0.086 | 0.109 | 0.072 | 0.083 |
| Fem | | | 0.219*** | 0.182** | 0.028 |
| Mas | | | | 0.085 | 0.112 |
| Fem × Sex | | | | | 0.236** |
| Mas × Sex | | | | | -0.059 |
| $R^2$ | 0.067*** | 0.074*** | 0.121*** | 0.125*** | 0.149*** |
| $\triangle R^2$ | | 0.007 | 0.046*** | 0.004 | 0.024* |

**Table 5. Hierarchical regression analysis of representational responding.**

| Model | β in Step 1 | β in Step 2 | β in Step 3 | β in Step 4 | β in Step 5 |
|---|---|---|---|---|---|
| Age | 0.072 | 0.056 | 0.046 | 0.043 | 0.045 |
| Ethnicity | 0.012 | 0.012 | 0.000 | -0.001 | 0.012 |
| MS | 0.130* | 0.132* | 0.120* | 0.118* | 0.126 |
| Sex | | 0.059 | 0.074 | 0.060 | 0.070 |
| Fem | | | 0.147* | 0.133* | -0.023 |
| Mas | | | | 0.033 | 0.053 |
| Fem × Sex | | | | | 0.237** |
| Mas × Sex | | | | | -0.050 |
| $R^2$ | 0.026 | 0.029 | 0.050* | 0.051* | 0.075** |
| $\triangle R^2$ | | 0.003 | 0.021* | 0.001 | 0.025* |

but it did have a strong effect on the motivational values for men ($t = 3.370$–$4.013$, $p < .001$). The interactions were plotted with the codes generated in Process (see Fig 2). In keeping with the results of the ANOVA, sex differences existed only in self-reported interest in infants and not in the motivational response towards infant faces.

**Table 6. Hierarchical regression analysis of evoked responding.**

| Model | β in Step 1 | β in Step 2 | β in Step 3 | β in Step 4 | β in Step 5 |
|---|---|---|---|---|---|
| Age | 0.039 | 0.015 | 0.002 | -0.005 | -0.003 |
| Ethnicity | 0.043 | 0.044 | 0.027 | 0.025 | 0.038 |
| MS | 0.059 | 0.062 | 0.046 | 0.042 | 0.050 |
| Sex | | 0.086 | 0.106 | 0.071 | 0.077 |
| Fem | | | 0.197** | 0.162* | -0.001 |
| Mas | | | | 0.081 | 0.068 |
| Fem × Sex | | | | | 0.242** |
| Mas × Sex | | | | | 0.003 |
| $R^2$ | 0.009 | 0.015 | 0.053** | 0.057** | 0.088** |
| $\triangle R^2$ | | 0.007 | 0.038** | 0.004 | 0.031** |

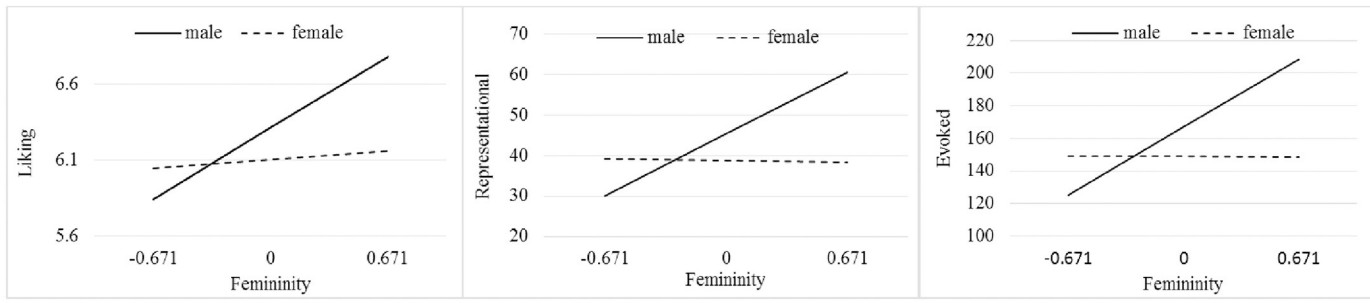

**Fig 2. The interactive effects on three domains of responses to infants.**

## Multilevel models

A random intercept multilevel regression with emotion nested within participants was used to examine the interaction between infant facial expressions and individual characteristic variables, specifically sex, gender roles, and the interactive effect of femininity and sex, in predicting motivation towards infants. Multilevel models were estimated with HLM software using maximum likelihood estimation.

Null models without explanatory variables accounted for 10.81% of the total variance in liking, 10.94% of the variance in representational responding, and 14.36% of the variance in evoked responding due to variation across participants. The emotion-level analysis demonstrated that facial expressions had a consistently significant effect on the three motivational values; specifically, the participants showed less liking and less representational and evoked responding to crying faces and more motivation towards laughing faces than neutral faces. After the level-1 predictors (i.e., emotion) were entered into the equation, the level-1 variance decreased by 68.0% for liking, 56.8% for representational responding, and 59.7% for evoked responding.

There were significant variations in the variance in the intercepts for liking ($\chi^2$ (299) = 1272.448, $p < .001$), representational responding ($\chi^2$ (299) = 947.499, $p < .001$), and evoked responding ($\chi^2$ (299) = 1114.460, $p < .001$), which allowed us to explore the effects of individual-level variables on these motivational values across emotions. The participant-level analysis did not reveal any significant interaction between emotion and gender roles or between emotion and the interaction between gender roles and sex across all the motivational values (Table 7). The variance in the random intercept was reduced by 15.5% for liking, 7.2% for representational responding, and 8.1% for evoked responding after the level-2 predictors were entered.

We did find a significant interaction between sex and laughing in liking but not in wanting. To clarify this effect, we separately conducted an independent t-test on liking of emotion. We found that men had higher liking than women for neutral infants ($t = -2.063$, $p < .05$, Cohen's $d = .238$), while both sexes showed equal liking for laughing and crying infants.

## Discussion

Social expectations about women's and men's social roles have undergone tremendous change. In China, as in other countries, women are increasingly expected to be financially independent, while men are more involved in parenting [46–48]. Against this background, the first goal of this study was to examine sex differences in China considering the mixed results of previous studies [12, 17–19, 21, 22]. Consistent with existing research [11–13], we found that Chinese women self-reported more interest in infants than men did.

**Table 7. Brief summary of 2-level multilevel models of motivation towards infant faces.**

| Fixed effects | Liking | | Representational | | Evoked | |
|---|---|---|---|---|---|---|
| | Estimate | *t* | Estimate | *t* | Estimate | *t* |
| The emotion level | | | | | | |
| Neutral | 6.342 | 70.136*** | 51.123 | 13.942*** | 169.610 | 19.682*** |
| Laughing | 1.418 | 20.945*** | 40.637 | 13.823*** | 106.540 | 14.684*** |
| Crying | -1.883 | -22.151*** | -70.210 | -17.931*** | -147.920 | -18.846*** |
| The participant level | | | | | | |
| Sex | 0.327 | 1.505 | 10.173 | 1.179 | 29.120 | 1.476 |
| Fem | 0.102 | 0.464 | 1.321 | 0.173 | 1.277 | 0.065 |
| Fem × Sex | 0.721 | 2.379* | 30.275 | 2.312* | 83.819 | 2.891*** |
| Laughing × Sex | -0.460 | -2.853** | -2.574 | -0.380 | -27.661 | -1.680 |
| Laughing × Fem | 0.099 | 0.661 | -0.968 | -0.194 | -1.062 | -0.067 |
| Laughing × Fem × Sex | -0.151 | -0.647 | -9.450 | -0.906 | -42.868 | -1.810 |
| Crying × Sex | 0.150 | 0.823 | -6.787 | -0.764 | -5.878 | -0.301 |
| Crying × Fem | -0.234 | -1.185 | -8.046 | -0.754 | -3.280 | -0.172 |
| Crying × Fem × Sex | 0.086 | 0.277 | -3.214 | -0.208 | -22.136 | -0.728 |

Note. To report the results in a concise way, only the estimates of the variables of interest were included in this table. The effects of other variables (i.e., age, marital status, ethnicity, masculinity and the interaction of masculinity and sex) are shown in the (S3 Table).

By contrast, no sex differences were found in any motivational values in response to infant faces irrespective of their facial expressions. Specifically, there were no differences between women and men in the extent of pleasure experienced, the efforts to prolong the viewing time of infants' faces, or the cognitive motivation to see the infants again. Furthermore, no interactive effects were found between sex and emotion in wanting, which means that regardless of the infants' emotions, women and men showed similar wanting for them. This finding corroborates the findings of existing studies [17–19, 22]. Nevertheless, the following question remains: why have some studies found stronger reactions to infant faces for women than for men [12, 23]?

Based on the results of the hierarchical regression analysis, our study found that femininity consistently and significantly predicted interest in infants and hedonic reactions to infants before the two-way interaction was entered. The results corroborate the findings of considerable research that has found that adults' internalization of femininity is positively associated with many facets of parenting, such as accessibility to children [31], expectations of parenting [10], and desire to have children [32].

We also found that femininity significantly interacted with sex in all motivational values towards infants (i.e., liking, representational responding, and evoked responding), which indicates that the effect of femininity on reactions to infant faces was more significant for men than for women. Put more bluntly, the more that men had internalized feminine traits, the stronger their liking and wanting for infants were, while women's responses to infant faces were less affected by their femininity. From a different perspective, we can say that men with high feminine traits have responses to infants that are comparable to and even higher than those of women, whereas men with low femininity show lower responses to infants than women do.

These results may indicate that acquired gender roles have an important effect on men's preferences for infant faces, which may explain why some previous studies found sex differences in responses to infant faces [12, 23], while other studies did not [10, 22]. That is, these

contradictory results emerged because the men included in previous studies had different levels of feminine traits. Moreover, the stable sex differences found in self-reported interest in infants may be due to the use of the questionnaire method, which is more susceptible to social desirability. In contrast, hedonic responses to infant faces are more authentic; thus, the difference between the sexes recorded in such tests becomes less significant.

Furthermore, multilevel models demonstrated that the three-way interaction was nonsignificant in liking and wanting infants, showing that the interactive effects between femininity and sex were equivalent for different facial expressions of infants. In other words, regardless of infants' emotions, feminine traits were more influential for men than for women in all components of motivational values towards infants. Combined with the lack of sex differences in terms of reactions to infants, the results may reflect that men's preferences for infants are mainly affected by socialization, while women, as primary caregivers, are more influenced by biological adaptation. In fact, the view of the plasticity of men's parental investment through socialization has also been supported by some studies that found that men's testosterone decreases when they become fathers [49, 50], and higher testosterone has been found to be associated with lower parental investment [51]. To some extent, these results are in line with social role theory and parental investment theory.

It is also interesting that we found that Chinese men internalized more masculinity than women but their internalization of femininity was similar to that of women. This result differs from the findings of many Western studies [10, 12, 31, 52, 53] that have found that women score higher on feminine traits than men. These results may be accounted for by cultural variability given the special background in China, where demographic and cultural values have changed greatly since the implementation of the one-child policy in the 1980s [54, 55]. The one-child policy prohibits Chinese men from having multiple offspring, which means they must allocate resources to increase offspring quality rather than quantity [56]. Furthermore, the imbalanced sex ratio caused by this policy [57] has led Chinese men to have to compete for a limited number of women. In other words, Chinese men may undertake more domestic activities than men in other countries, and based on social role theory [30], the associated skills, values, and motives are incorporated into their feminine traits, but further research is required to test this hypothesis. For example, future studies could include samples from different countries or from different stages of development in China (e.g., before the implementation of the one-child policy, during the policy, and after the end of the policy) and then compare the differences in the social roles men assume and gender roles.

We also observed a main effect of emotion on adults' behavioral responses to infant faces. Adults demonstrated more liking and wanting for infants with positive emotions than for neutral and crying infants and less liking and wanting for infants with negative emotions than for laughing and neutral infants. This finding is also consistent with those of previous studies [20, 37–39]. Adults were found to rate smiling and neutral children as cuter, more adoptable, and less distressing than crying children and viewed videos of smiling children for longer durations than videos of crying children [20]. Neuroimaging studies have also found that regions of brain activity can be differentiated by infant expressions [37, 38]. This result demonstrates that infant facial expressions are important and should be considered in future studies.

Despite the contribution of this study, a number of specific limitations need to be considered. First, this study is based on a correlational design, and no inferences of causality can be drawn. Future studies regarding the nature of these relationships should adopt a longitudinal method to verify the results. Second, interest and motivation responses to infant faces differ from caregiving behaviors. Thus, the relationship between femininity and parenting needs to be investigated in experiments with better ecological validity. Finally, our study was conducted in a different social environment from those of most existing Western studies. In the future, a

cross-cultural comparative study should be conducted to provide a clearer picture of the effects of the social environment.

## Conclusion

This study employed a relatively large sample of adults to extend the understanding of sex differences and the role of gender role orientation with respect to parenting among Chinese adults using the questionnaire and behavioral paradigm methods. First, this study helps to explain the inconsistency in existing studies with respect to responses to infants. Future studies should consider not only biological sex differences in parenting but also the internalization of gender roles. Second, we found men's feminine traits to be more influential than women's on their responses to infant faces. This result represents, to some extent, the malleability of men's parenting roles. These findings help deepen the understanding of parental investment theory and social role theory. Third, this study confirmed the effect of infants' facial expressions on the hedonic responses of adults. Finally, we found that, perhaps due to the special social background in China, men's feminine traits were comparable to women's traits, which may be due to cultural variability.

## Supporting information

**S1 Table. Sample size broken down by sex, marital status and ethnicity.**
(DOCX)

**S2 Table. Details of ANOVA.**
(DOCX)

**S3 Table. Full summary of 2-level multilevel models of motivation towards infant faces.**
(DOCX)

**S1 Data.**
(RAR)

## Author Contributions

**Conceptualization:** Gang Cheng, Yuncheng Jia, Wen Zhang, Dajun Zhang.

**Data curation:** Fangyuan Ding, Gang Cheng, Wen Zhang, Nan Lin, Wenjing Mo.

**Formal analysis:** Fangyuan Ding, Yuncheng Jia.

**Funding acquisition:** Gang Cheng, Dajun Zhang.

**Investigation:** Fangyuan Ding, Yuncheng Jia, Wen Zhang, Nan Lin, Wenjing Mo.

**Methodology:** Fangyuan Ding, Yuncheng Jia, Nan Lin, Wenjing Mo.

**Project administration:** Dajun Zhang.

**Resources:** Gang Cheng.

**Supervision:** Gang Cheng, Dajun Zhang.

**Visualization:** Fangyuan Ding.

**Writing – original draft:** Fangyuan Ding.

**Writing – review & editing:** Fangyuan Ding, Gang Cheng, Dajun Zhang.

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
