## [Decision Letter · Decision Letter 0]

31 Jul 2020

PONE-D-20-17379

The role of sex and femininity in preferences for unfamiliar infants among chinese adults

PLOS ONE

Dear Dr. Dajun Zhang,

Thank you for submitting your manuscript to PLOS ONE. After careful consideration, we feel that it has merit but does not fully meet PLOS ONE’s publication criteria as it currently stands. Therefore, we invite you to submit a revised version of the manuscript that addresses the points raised during the review process.

We look forward to receiving your revised manuscript.

Kind regards,

Yuka Kotozaki

Academic Editor

PLOS ONE

Journal Requirements:

3.Thank you for including your ethics statement:  "the Ethics Committee of Southwest University (No. 2014179)".   

Please amend your current ethics statement to confirm that your named institutional review board or ethics committee specifically approved this study.

4.We note that Figure(s) [1] in your submission contain copyrighted images. All PLOS content is published under the Creative Commons Attribution License (CC BY 4.0), which means that the manuscript, images, and Supporting Information files will be freely available online, and any third party is permitted to access, download, copy, distribute, and use these materials in any way, even commercially, with proper attribution. For more information, see our copyright guidelines: http://journals.plos.org/plosone/s/licenses-and-copyright.

1.    You may seek permission from the original copyright holder of Figure(s) [1] to publish the content specifically under the CC BY 4.0 license.

5.Thank you for stating the following financial disclosure:

 [The funders had no role in study design, data collection and analysis, decision to publish, or preparation of the manuscript.].

Reviewers' comments:

Reviewer's Responses to Questions

**Comments to the Author**

1. Is the manuscript technically sound, and do the data support the conclusions?

Reviewer #1: Yes

Reviewer #2: Yes

2. Has the statistical analysis been performed appropriately and rigorously? 

Reviewer #1: Yes

Reviewer #2: Yes

3. Have the authors made all data underlying the findings in their manuscript fully available?

Reviewer #1: Yes

Reviewer #2: No

4. Is the manuscript presented in an intelligible fashion and written in standard English?

Reviewer #1: Yes

Reviewer #2: Yes

5. Review Comments to the Author

Reviewer #1: 1. Do not repeat and remove “We assessed this question using a self-reported method and behavioral responses to infant faces” (p.13, line 119)

2. When recruiting the subjects, was AGE (e.g., 18 to 40) an inclusion criterion because the focus of this study was about infant preferences among childless adults. If not, it would be hard to believe no one in this sample was above 40 years of age. If this is an inclusion criterion, then this study would be about infant preference among child-bearing age adults.

3. You describe the subjects being “healthy childless adults” (p.13). Does it mean that health status and childless were two of the inclusion criteria? How do you determine they were healthy?

4. Remove “to participate” on line 134 (p.14). Informed consent is always for participation; to remove these two words so that it does not seem to be reductant with the word participants in the same sentence.

5. The term “adult women” should be just “women” which already implies adults; otherwise, they are girls, children, or adolescents.

6. Method: The research questions need rewording to show the “sex differential” variable is about the childless adults, not about the infants’ gender preference. It was clear after reading the results but not clear at the beginning. If this is a research question, it must be reworded to something like this: This study seems to have only one major question: “To what extent do gender and gender role orientation among childless adults influence their infant preferences?” The manipulation of providing laughing and crying facial expressions is a method to check if the findings to the answer to this question may change. After the research question, you may add a question with a procedure of manipulation, e.g., “In this study, a manipulation procedure has been added to test the following: Would a change in the infant’s emotion (from a neutral facial expression to laughing or crying) change these childless adults’ original infant preferences?”

7. One finding: “It is also interesting that we found that Chinese men internalized more masculinity than women but showed similar femininity internalization to that of women” What does it really imply? It sounds that women have become less feminine and men become multi-oriented in their gender roles? Wonder how the feminine characteristics measure fits well with young adults vs. those in their 30s and 40s because the educational and socialization process between these two groups of adults may be very different due to the economic and cultural shift with China’s economy growth in the last two decades. While age does not seem to correlate with femininity, it is significantly correlated with masculinity, and femininity and masculinity are also significantly correlated. Would men on the other hand tend to be more feminine?

8. Additional editorial support is needed. E.g., do not start a new paragraph with “However”.

9. Interesting findings but not very clear about how the feminine vs masculine characteristics are internalized by men (if this is a reason to further explore the impact of the femininity factor.

Reviewer #2: This is a novel study on the effect of gender role on infant face preferences. Motivated by the parental investment theory and social role theory, the authors hypothesize that instead of dichotomic difference between the two sexes with respect to preferences to infant faces, there might exist more gradient difference in viewing infant faces due to gender role orientation. Combining questionnaire and behavioral tests, the authors found that gender role affects males’ responses more than females. I found that article well written, the analysis sound, and the results interesting. I only have very minor suggestions for the authors to incorporate into their revision before this work is published.

1. The introduction needs to mention previous research on infant face preferences by the Chinese population, and how the Chinse participants are expected to behave similarly or differently given what we have known about the cross-cultural differences. The discussion covered some of these points, but it will be very helpful that the context is properly laid out at in the introduction.

2. Likewise, the introduction needs to discuss ethnic composition of the Chinese population and its potential impact before mentioning of the inclusion of different ethnic groups in the methods.

3. Page 6, the last line, the period is missing.

6. PLOS authors have the option to publish the peer review history of their article (what does this mean?). If published, this will include your full peer review and any attached files.

Reviewer #1: **Yes: **Monit Cheung, PhD

Reviewer #2: No

---

## [Author Response · Author response to Decision Letter 0]

25 Aug 2020

Journal Requirements:

Reply: Thank you. We have tried our best to format our manuscript according to the templates. If there are any omissions, please point them out. We will continue to make corrections.

Reply: Thank you for pointing out our problems. We have including the title page in the manuscript.

3. Thank you for including your ethics statement: "the Ethics Committee of Southwest University (No. 2014179)". 

Please amend your current ethics statement to confirm that your named institutional review board or ethics committee specifically approved this study.

Reply: Thanks for your suggestion. We have revised it in our revised manuscript (see p.7, line 156). We also add the statement to the submission form.

4. We note that Figure(s) [1] in your submission contain copyrighted images. All PLOS content is published under the Creative Commons Attribution License (CC BY 4.0), which means that the manuscript, images, and Supporting Information files will be freely available online, and any third party is permitted to access, download, copy, distribute, and use these materials in any way, even commercially, with proper attribution. For more information, see our copyright guidelines: http://journals.plos.org/plosone/s/licenses-and-copyright. 

1. You may seek permission from the original copyright holder of Figure(s) [1] to publish the content specifically under the CC BY 4.0 license.

Reply: Thanks for your detailed guidance. We have obtained the written permission and submitted it as an “other” file. Besides, we also claimed this below our Fig.1.

5.Thank you for stating the following financial disclosure:

 [The funders had no role in study design, data collection and analysis, decision to publish, or preparation of the manuscript.].

a. Please clarify the sources of funding (financial or material support) for your study. List the grants or organizations that supported your study, including funding received from your institution.

d. If you did not receive any funding for this study, please state: “The authors received no specific funding for this work.”

Reply: Thanks for pointing out our problems. We have included our amended statements within our cover letter (see the blue paragraph).

Review Comments to the Author

Reviewer #1: 1. Do not repeat and remove “We assessed this question using a self-reported method and behavioral responses to infant faces” (p.13, line 119)

Reply: Thank you for your suggestion. I have deleted the redundant sentence.

2. When recruiting the subjects, was AGE (e.g., 18 to 40) an inclusion criterion because the focus of this study was about infant preferences among childless adults. If not, it would be hard to believe no one in this sample was above 40 years of age. If this is an inclusion criterion, then this study would be about infant preference among child-bearing age adults.

Reply: Thanks for your valuable comment. Your guess is right. We aimed to recruit child-bearing age childless adults at the beginning. Thus, we chose 18-40 childless adults. I have added this in Participants section (see p.7, line 151).

3. You describe the subjects being “healthy childless adults” (p.13). Does it mean that health status and childless were two of the inclusion criteria? How do you determine they were healthy?

Reply: Thanks a lot for pointing out our problems. The health status in this study mainly means no history of mental illness. We have added these recruiting criteria in Participants section (see p.7, line 150).

4. Remove “to participate” on line 134 (p.14). Informed consent is always for participation; to remove these two words so that it does not seem to be reductant with the word participants in the same sentence.

Reply: Thank you for your patient instruction. I deleted it as your guidance (see p.7, line 159).

5. The term “adult women” should be just “women” which already implies adults; otherwise, they are girls, children, or adolescents.

Reply: Thanks for teaching us. I deleted “adult” before women throughout the paper.

6. Method: The research questions need rewording to show the “sex differential” variable is about the childless adults, not about the infants’ gender preference. It was clear after reading the results but not clear at the beginning. If this is a research question, it must be reworded to something like this: This study seems to have only one major question: “To what extent do gender and gender role orientation among childless adults influence their infant preferences?” The manipulation of providing laughing and crying facial expressions is a method to check if the findings to the answer to this question may change. After the research question, you may add a question with a procedure of manipulation, e.g., “In this study, a manipulation procedure has been added to test the following: Would a change in the infant’s emotion (from a neutral facial expression to laughing or crying) change these childless adults’ original infant preferences?”

Reply: Thanks for your valuable suggestions. Following your comment, we combined the first two research questions (a & b) and rephrased them into the first research question you offered Then the second research question you help us rewording here is identical to what we want to test at first. However, the example you offering: “Would a change in the infant’s emotion (from a neutral facial expression to laughing or crying) change these childless adults’ original infant preferences?” seems inconsistent with our research question. Thus, based on your suggestions, we finally choose to keep two research questions: one is the main research question you provided to us; the other is whether the effect in question 1 will be affected by infants’ emotions (see p.6, line 142).

7. One finding: “It is also interesting that we found that Chinese men internalized more masculinity than women but showed similar femininity internalization to that of women” What does it really imply? It sounds that women have become less feminine and men become multi-oriented in their gender roles? Wonder how the feminine characteristics measure fits well with young adults vs. those in their 30s and 40s because the educational and socialization process between these two groups of adults may be very different due to the economic and cultural shift with China’s economy growth in the last two decades. While age does not seem to correlate with femininity, it is significantly correlated with masculinity, and femininity and masculinity are also significantly correlated. Would men on the other hand tend to be more feminine?

Reply: Thanks a lot for your interesting ideas. In our discussion, we are trying to explain why Western women have higher femininity than men but among Chinese adults, there are no sex differences. We account for this to cultural differences. It is possible that, we think, Chinese men are becoming more feminine than western men (e.g, tender) due to the one-child policy or imbalanced sex ratio in China, which is different from western countries. Future cross-cultural studies will verify this.

Then, what you are focused on here is about the cultural change in China and how the economic and cultural shift in China affects Chinese adults’ gender roles. We feel the research question is very interesting but beyond the scope of this study because almost all of our samples (99.3%) were born after the reform and opening up in China. It is worth studying in the future by recruiting individuals from different stages of development in China, such as individuals before the implementation of the one-child policy and those after the two-child policy or individuals from economically developed and underdeveloped regions (China’s economic growth in the last two decades was uneven across regions). We have added this idea to the prospect of future studies in discussion (see p.20, line 372).

8. Additional editorial support is needed. E.g., do not start a new paragraph with “However”.

Reply: Thanks for your comment. In this revision of manuscript, we asked for help for the language polishing from AJE Company.

9. Interesting findings but not very clear about how the feminine vs masculine characteristics are internalized by men (if this is a reason to further explore the impact of the femininity factor.

Reply: Thank you for your comment. Based on social role theory, to the extent that women and men occupy roles involving domestic activities or economically productive activities, the associated skills, values, and motives are incorporated into their gender roles. We have introduced it in the introduction but we found it is unclear in our discussion. So inspired by you, we revised it a little in that part (see p.20, line369).

Reviewer #2: This is a novel study on the effect of gender role on infant face preferences. Motivated by the parental investment theory and social role theory, the authors hypothesize that instead of dichotomic difference between the two sexes with respect to preferences to infant faces, there might exist more gradient difference in viewing infant faces due to gender role orientation. Combining questionnaire and behavioral tests, the authors found that gender role affects males’ responses more than females. I found that article well written, the analysis sound, and the results interesting. I only have very minor suggestions for the authors to incorporate into their revision before this work is published.

Reply: Thank you for your kind comments. All the revisions were marked in blue. 

1. The introduction needs to mention previous research on infant face preferences by the Chinese population, and how the Chinse participants are expected to behave similarly or differently given what we have known about the cross-cultural differences. The discussion covered some of these points, but it will be very helpful that the context is properly laid out at in the introduction.

Reply: Thank you for your suggestions. Relatively few studies examined sex differences in preferences in the Chinese sample. In our review, two studies among Chinese found no sex differences in behavioral responses and brain activity, one of which found women report more interest in infants. So basically the findings among China are similar to many studies in western countries. In the discussion section, the cultural differences between China and Western countries are mainly used to explain the sex differences in femininity. Studies in western countries reported women with higher feminine traits than men while in our sample, men showed similar levels of feminine traits with women. This difference is not our main research question. Thus, based on your suggestion, we only add a little about previous research on infant preferences among Chinese in introduction (see p.6, line 138), but didn’t add the introduction of cultural differences.

2. Likewise, the introduction needs to discuss ethnic composition of the Chinese population and its potential impact before mentioning of the inclusion of different ethnic groups in the methods.

Reply: Thanks for your suggestions. Following your suggestions, I introduced a little ethnic composition in China and discuss the different fertility policies between Han and minority groups (see p.6, line 126).

3. Page 6, the last line, the period is missing.

Reply: Thank you for pointing our carelessness. We have added it and the full text was checked in case there were any similar mistakes.

---

## [Decision Letter · Decision Letter 1]

29 Oct 2020

The role of sex and femininity in preferences for unfamiliar infants among chinese adults

PONE-D-20-17379R1

Dear Dr. Dajun Zhang,

We’re pleased to inform you that your manuscript has been judged scientifically suitable for publication and will be formally accepted for publication once it meets all outstanding technical requirements.

Kind regards,

Yuka Kotozaki

Academic Editor

PLOS ONE

Additional Editor Comments (optional):

Reviewers' comments:

Reviewer's Responses to Questions

**Comments to the Author**

1. If the authors have adequately addressed your comments raised in a previous round of review and you feel that this manuscript is now acceptable for publication, you may indicate that here to bypass the “Comments to the Author” section, enter your conflict of interest statement in the “Confidential to Editor” section, and submit your "Accept" recommendation.

Reviewer #1: All comments have been addressed

Reviewer #2: All comments have been addressed

2. Is the manuscript technically sound, and do the data support the conclusions?

Reviewer #1: Yes

Reviewer #2: Yes

3. Has the statistical analysis been performed appropriately and rigorously? 

Reviewer #1: Yes

Reviewer #2: Yes

4. Have the authors made all data underlying the findings in their manuscript fully available?

Reviewer #1: Yes

Reviewer #2: Yes

5. Is the manuscript presented in an intelligible fashion and written in standard English?

Reviewer #1: Yes

Reviewer #2: Yes

6. Review Comments to the Author

Reviewer #1: All comments are addressed or placed in future study recommendations. This article has some unique aspects on femininity that may have been affected by the one-child policy. While the two-child policy is in effect, the previously policy has deeply affected current adults' mind toward the gender of children.

Reviewer #2: All my previous comments have been adequately addressed. I therefore recommend publication of this article.

7. PLOS authors have the option to publish the peer review history of their article (what does this mean?). If published, this will include your full peer review and any attached files.

Reviewer #1: **Yes: **Monit Cheung

Reviewer #2: **Yes: **Fangfang Li

---

## [Editor Report · Acceptance letter]

3 Nov 2020

PONE-D-20-17379R1 

The role of sex and femininity in preferences for unfamiliar infants among Chinese adults 

Dear Dr. Zhang:

I'm pleased to inform you that your manuscript has been deemed suitable for publication in PLOS ONE. Congratulations! Your manuscript is now with our production department. 

Kind regards, 

on behalf of

Dr. Yuka Kotozaki 

Academic Editor

PLOS ONE